# Iron Uptake Controls *Trypanosoma cruzi* Metabolic Shift and Cell Proliferation

**DOI:** 10.3390/antiox12050984

**Published:** 2023-04-22

**Authors:** Claudia F. Dick, Carolina L. Alcantara, Luiz F. Carvalho-Kelly, Marco Antonio Lacerda-Abreu, Narcisa L. Cunha-e-Silva, José R. Meyer-Fernandes, Adalberto Vieyra

**Affiliations:** 1Instituto de Biofísica Carlos Chagas Filho, Universidade Federal do Rio de Janeiro, Rio de Janeiro 21941-902, RJ, Brazil; alcantara@biof.ufrj.br (C.L.A.); narcisa@biof.ufrj.br (N.L.C.-e.-S.); avieyra@biof.ufrj.br (A.V.); 2Centro Nacional de Biologia Estrutural e Bioimagem, Universidade Federal do Rio de Janeiro/CENABIO, Rio de Janeiro 21941-902, RJ, Brazil; 3Instituto de Bioquímica Médica Leopoldo de Meis, Universidade Federal do Rio de Janeiro, Rio de Janeiro 21941-902, RJ, Brazil; lfkelly@bioqmed.ufrj.br (L.F.C.-K.); mantonio.abreu@bioqmed.ufrj.br (M.A.L.-A.); meyer@bioqmed.ufrj.br (J.R.M.-F.); 4Programa de Pós-Graduação em Biomedicina Translacional /BIOTRANS, Universidade do Grande Rio, Duque de Caxias 25071-202, RJ, Brazil

**Keywords:** trypanosomatids, growth and differentiation, parasite oxidative stress, mitochondrial function, ATP synthesis, parasite lipid content

## Abstract

(1) Background: Ionic transport in *Trypanosoma cruzi* is the object of intense studies. *T. cruzi* expresses a Fe-reductase (TcFR) and a Fe transporter (TcIT). We investigated the effect of Fe depletion and Fe supplementation on different structures and functions of *T. cruzi* epimastigotes in culture. (2) Methods: We investigated growth and metacyclogenesis, variations of intracellular Fe, endocytosis of transferrin, hemoglobin, and albumin by cell cytometry, structural changes of organelles by transmission electron microscopy, O_2_ consumption by oximetry, mitochondrial membrane potential measuring JC-1 fluorescence at different wavelengths, intracellular ATP by bioluminescence, succinate-cytochrome c oxidoreductase following reduction of ferricytochrome c, production of H_2_O_2_ following oxidation of the Amplex^®^ red probe, superoxide dismutase (SOD) activity following the reduction of nitroblue tetrazolium, expression of SOD, elements of the protein kinase A (PKA) signaling, TcFR and TcIT by quantitative PCR, PKA activity by luminescence, glyceraldehyde-3-phosphate dehydrogenase abundance and activity by Western blotting and NAD^+^ reduction, and glucokinase activity recording NADP^+^ reduction. (3) Results: Fe depletion increased oxidative stress, inhibited mitochondrial function and ATP formation, increased lipid accumulation in the reservosomes, and inhibited differentiation toward trypomastigotes, with the simultaneous metabolic shift from respiration to glycolysis. (4) Conclusion: The processes modulated for ionic Fe provide energy for the *T. cruzi* life cycle and the propagation of Chagas disease.

## 1. Introduction

The etiological agent of Chagas disease, *Trypanosoma cruzi*, has a complicated life cycle that alternates between intermediate invertebrates and definitive mammal hosts [1], and Chagas disease is considered a neglected disease worldwide [2,3].

Iron (Fe) is one nutritional element that controls *T. cruzi* growth and differentiation during its life cycle since it is a necessary micronutrient for all forms of life and a cofactor of many enzymes in a considerable number of metabolic pathways [4]. Fe is also hazardous because of its potential to accelerate the creation of reactive oxygen species (ROS), and all biological systems have evolved mechanisms for managing Fe intake, metabolism, and storage [5].

Fe is essentially physiologically inaccessible due to the limited solubility of its thermodynamically stable +3 oxidation state in the presence of O_2_ at neutral pH [6,7]. The concentration of free Fe in the environment ranges between 10^−9^ and 10^−18^ M, which is lower than the concentration necessary for microbial development [8]. Fe is required for DNA synthesis [9,10], energy generation [11], and oxidative stress in trypanosomes [12]. Furthermore, mammalian hosts sequester free Fe into proteins such as transferrin and lactoferrin [13,14,15], resulting in a free Fe concentration in serum of roughly 10^−24^ M [13,14,15]. Thus, there are three essential sources of Fe in mammals’ bodies that a pathogenic microbe may use: (i) transferrin, (ii) ferritin, and (iii) heme-containing proteins like hemoglobin [16].

*Trypanosoma cruzi* requires iron (Fe) for growth, in vitro proliferation of epimastigotes forms (mobilizing heminic or non-heminic Fe), and pathogenicity in mice [17]. *Trypanosoma cruzi* can hijack Fe-proteins from mammalian hosts. In culture, adding deferoxamine, a Fe chelator, or transferrin-free serum can reduce amastigotes cell multiplication, demonstrating that Fe is an essential nutrient [18]. This parasite has evolved human transferrin receptors that bind exogenous transferrin. Acid treatment does not remove transferrin attached to amastigote cells, indicating that this transferrin may be internalized and used [18]. Transferrin is taken up by the cytostome, a specialized structure consisting of a profound membrane invagination in the anterior area near the flagellar pocket [19]. *Trypanosoma cruzi* also uses heme as a Fe source; it can boost *T. cruzi* growth in culture in a dose-dependent way [20]. Furthermore, *T. cruzi* epimastigotes internalize heme/porphyrin through a process that might be mediated by an ABC transporter protein [21]. However, because no heme oxidase gene is indicated on the *T. cruzi* genome, the first heminic ring hydrolysis for Fe release is the limiting step for pathogenic trypanosomatids using heme [5].

Since Fe is present in aerobic conditions as Fe^3+^, it must be converted to Fe^2+^ by the Fe-reductase enzyme [22] to be transported across the plasma membrane. There is much evidence that Fe^3+^ reduction is frequently linked to Fe^2+^ transport in bacteria, yeast, plant, and animal cells [23]. As a result, the identification of Fe-reductase activity in *Leishmania chagasi* [24], *L. amazonensis* [22], and, subsequently, *T. cruzi* [25] was a strong signal of the presence of a Fe^2+^ transport mechanism in trypanosomatids.

The discovery of Fe-reductase activity in trypanosomatids reinforced the hypothesis of a two-step Fe transport mechanism: first, the reduction of Fe^3+^ to Fe^2+^, followed by the absorption of Fe^2+^ by specialized transporters [26]. In addition, the finding of a Fe transporter (LIT, [27]) in the plasma membrane of *L. amazonensis*, which belongs to the zinc and iron transporter family (ZIP family), gave early support to this idea. ZIP family members are said to be capable of transporting Zn^2+^; however, some members of this family can also transport Fe^2+^ [28]. Recently, the discovery of the TcIT Fe transporter in *T. cruzi* [29] supports the hypothesis of a functional link between TcFR [25] and TcIT for Fe^2+^ uptake in this parasite.

Due to its low redox potential, Fe is a suitable element for redox catalysis processes [30,31], acting as an electron donor and receptor and being able to catalyze the formation of Once internalized, free Fe must be stored or processed as it enters the cytosol to avoid the generation of ROC. Corrêa et al. [32] discovered Fe in the acidocalcisomes of *T. cruzi* blood trypomastigotes. A putative Fe transporter in the form of a metal ion in the acidocalcisome of *T. cruzi* supports the hypothesis of Fe storage in this organelle [33]. This metabolic element has lately gained prominence due to the discovery that Fe mobilization and oxidative stress play a critical role in the parasite’s persistence and survival in the tissues of the mammalian host—a role that had not previously been proven [34].

Although Fe is a critical micronutrient for trypanosomatids, as previously stated [9,10,11,12], there are still many unknowns about its involvement in the life cycle and pathogenicity of these organisms, as well as the ways in which it is acquired and used. The main aim of the present work was to investigate whether exogenous ionic Fe modulates *T. cruzi’s* redox status and metabolic pathways, as well as the proliferation and differentiation of the parasite. The study revealed molecular mechanisms and intracellular processes modulated by exogenous ionic Fe that had not previously been reported.

## 2. Materials and Methods

### 2.1. Epimastigote Growth and Metacyclogenesis

*Trypanosoma cruzi* (Dm28c strain) epimastigotes were grown in stationary phase at 28 °C in Brain Heart Infusion (BHI) medium supplemented with 10% FBS, 30 µM hemin, and 1% penicillin-streptomycin (P/S) cocktail (referred to hereafter as Regular Media, RM). Iron-Depleted Media (IDM) was prepared using an iron-free BHI medium, as described by Dick et al. [25]. Briefly, BHI medium without hemin addition was treated with Chelex (5 g/100 mL) for 1 h at room temperature and sterilized by filtration using 0.22 µm pore-size filters. To this Iron-Free BHI medium was added 1% P/S cocktail and 10% Iron-Free FBS. The iron-free FBS was prepared by adding 10 mM ascorbic acid for 6–7 h at 37 °C until the optical density at 405 nm had decreased by 50%. Then, the solution was supplied with 5 g of Chelex resin per 100 mL and incubated at room temperature under stirring at 50 rpm for 3–4 h, filtered to remove the resin, and dialyzed (with a cutoff of 2000 Da) against 4 l of cold, sterile PBS for 6 h, changing the solution every 2 h. The iron-free FBS was also sterilized by filtration using 0.22 µm pore-size filters and stored at −20 °C. Iron-Free FBS Iron-Depleted Media with Fe (IDM + Fe) was made the same way as IDM but with 8 µM Fe-citrate added. The parasites were inoculated (10^6^ cells/mL) into the BHI medium on the sixth day of growth to test epimastigotes proliferation (RM, IDM, or IDM + Fe). Every day, cell proliferation was measured by counting the number of cells in a hemocytometer. Dm28c is a strain that differentiates “in vitro” and was previously used in our previous studies that demonstrated the existence of a Fe-reductase and a Fe-transporter in *T. cruzi* [25,29]. This strain is deposited in the Collection of Trypanosoma from Wild and Domestic Mammals and Vectors (COLTRYP), Oswaldo Cruz Foundation, Rio de Janeiro, Brazil.

Metacyclogenesis was induced according to Koeller et al. [35]. Epimastigotes in the transition from the logarithmic to the stationary phase were adjusted to 5 × 10^8^ parasites/mL in triatomine artificial urine (TAU) medium (190 mM NaCl, 17 mM KCl, 2 mM MgCl_2_, 2 mM CaCl_2_, 0.035% (*w*/*v*) NaHCO_3_, and 8 mM phosphate buffer at pH 6.0). After 2 h at 28 °C, the cultures were diluted 100-fold in 10 mL TAU medium supplemented with 10 mM proline and 250 mM glucose (TAU-P) plus 500 g/mL G418 (Sigma-Aldrich, Saint Louis, MO, USA) and transferred to T25 flasks—lying at a 45° angle to increase the area in contact with O_2_—and kept at 28 °C to promote metacyclogenesis. Following 3–5 days, parasites were counted using hemocytometry, and the proportion of metacyclic epimastigotes (Tryp) was determined using their morphology after Giemsa staining.

### 2.2. Intracellular Fe Concentration Determination

A colorimetric test based on ferrozine was used to assess the quantity of intracellular ionic Fe accumulated under different circumstances, as described before [29]. Briefly, suspensions containing 10^8^ parasites were obtained from various cultures and washed three times with PBS pretreated with 5 g/100 mL Chelex resin (Sigma-Aldrich). The cells were lysed with 100 µL of 50 mM NaOH, and then 100 µL of 10 mM HCl was added; the release of ionic Fe bound to intracellular structures was induced by adding 100 µL of a mixture containing 1.4 M HCl and 4.5% (*w*/*v*) KMnO_4_ (1:1) to the cell lysate, followed by incubation at 60 °C for 2 h. Then, 30 µL Fe detection reagent was added (6.5 mM ferrozine, 6.5 mM neocuproin, 2.5 M ammonium acetate, and 1 M ascorbic acid). The sample’s absorbance at 550 nm was measured after 30 min of incubation at room temperature. A standard curve with known FeCl_3_ concentrations (0–75 µM; [36]) (Merck, Darmstadt, Germany) was used to calculate the Fe content.

### 2.3. Real-Time-PCR

Total *T. cruzi* RNA was extracted using a Direct-zol RNA Miniprep Kit (Zymo Research, Orange, CA, USA) from epimastigotes kept at RM, IDM, or IDM + Fe for 5 days (as indicated in the figure legends). The high-capacity cDNA reverse transcription kit was used to reverse-transcribe whole RNA (Thermo Fisher Scientific, Waltham, MA, USA). For RT-PCR, 100 ng/µL cDNA per well (15 µL total volume) was utilized, coupled with a 5 µM primer mix and a 7 µL PowerUp SYBR green master solution (Thermo Fisher Scientific). Primers were designed using the Primer3 software, with predicted amplicon sizes of 100 pb each [37]. The primers for amplification are shown in Table 1. Gene expression data were normalized to an endogenous reference, β-tubulin. The expression ratios were determined using the threshold cycle (ΔΔCT) [38].

### 2.4. Endocytosis Assay

Epimastigotes were submitted to endocytosis with 30 µg/mL of transferrin-FITC, hemoglobin-FITC, or BSA-FITC in Roswell Park Memorial Institute (RPMI) medium for 30 min at 28 °C. The parasites were fixed with 4% (*v*/*v*) paraformaldehyde in phosphate-buffered saline (PBS, pH 7.2) for 1 h for flow cytometry and imaging using the fluorescence microscope AxioObserver (Zeiss, Oberkochen, Germany) after staining with DAPI.

### 2.5. Cell Cytometry

After the endocytosis assay, tracer uptake was measured on a BD Accuri C6 flow cytometer (Becton Dickinson Bioscience, BDB, San José, CA, USA), counting 10,000 events at the FL2 channel. The data were analyzed by the BD Accuri C6 software. This analysis was performed in three independent experiments.

### 2.6. Transmission Electron Microscopy

Samples were fixed with 2.5% (*v*/*v*) glutaraldehyde in 0.1 M cacodylate buffer (pH 7.2) for 1 h at room temperature. After a wash in cacodylate buffer, cells were post-fixed using an osmium-thiocarbohydrazide-osmium (OTO) protocol as already described [39]. Samples were washed in water, dehydrated in an acetone series, and embedded in epoxy resin (EMBED 812 resins, EMS, Hatfield, PA, USA). Ultrathin sections were cut with a UC7 ultramicrotome (Leica, Wetzlar, Germany), stained with 5% (*w*/*v*) uranyl acetate and lead citrate, and observed in an HT7800 transmission electron microscope (Hitachi, Tokyo, Japan) operating at 100 kV.

### 2.7. Protein Kinase A (PKA) Activity

Epimastigotes cells (5 × 10^7^ cells/mL) were washed twice in ice-cold phosphate buffer saline (PBS, pH 7.2) and lysed in 0.5 mL radioimmunoprecipitation assay buffer (RIPA buffer) for 30 min. PKA activity was assayed in the presence of 4 mM Hepes-Tris (pH 7.0), 0.4 mM MgCl_2_, 1 mM CaCl_2_, 1 µM ATP, and 50 µg of lysed cells in a final volume of 50 µL, in the presence or absence of 5 µM 3-isobutyl-1-methylxanthine (IBMX, a PKA activator), in MTS-11C mini tubes (Axygen Scientific, Union City, CA, USA). The reaction was triggered by adding 50 μL of the Kinase-Glo luminescent kit, and after 10 min at 37 °C, the samples were placed in a GloMax Multi JR detection system (Promega Corporation, Fitchburg, WI, USA). PKA activity was quantified as the difference between the reading in the presence of IBMX and in the absence of the activator.

### 2.8. High-Resolution Respirometry in Different Respiratory States

Oxygen consumption of intact epimastigotes (5 × 10^7^ parasites/chamber) was measured using an O2k-system high-resolution oxygraph (Oxygraph-2K; Oroboros Instruments, Innsbruck, Austria) at 28 °C with continuous stirring. The cells were suspended in a 2 mL respiration solution containing 100 mM sucrose, 50 mM KCl, and 50 mM Tris–HCl (pH 7.2), and 50 μM digitonin was added to permeabilize the parasites. Following that, 10 mM succinate and 200 μM ADP were added. Uncoupled respiration was induced with 3 μM carbonyl cyanide 4-(trifluoromethoxy)phenylhydrazone (FCCP), and then blocked with 2.5 μg/mL antimycin A to assess residual O_2_ consumption [29]. Oxygen concentrations and O_2_ consumption were recorded using DatLab software coupled to Oxygraph-2K.

### 2.9. Succinate-Cytochrome C Oxidoreductase Activity

The activity of succinate-cytochrome c oxidoreductase (complex II/III) was determined by the increase in absorbance due to the reduction of ferricytochrome c at 550 nm [40,41]. Frozen-thawed parasite homogenates (50 µg) were incubated with 25 mM potassium phosphate (pH 7.4), 10 mM succinate, 1 mM KCN, and 5 mM MgCl_2_ for 20 min to allow for complete activation of succinate dehydrogenase, after which the reaction was initiated with 50 µM horse heart cytochrome c and monitored for 2 min. Protein concentrations were determined by the Bradford method, using bovine serum albumin as a standard [42].

### 2.10. Mitochondrial Membrane Potential

Mitochondrial membrane potential was analyzed in the *T. cruzi* epimastigotes using the MitoProbe JC-1 assay kit (Molecular Probes; Thermo Fisher Scientific). Epimastigotes (1 × 10^7^ cells/mL) were loaded with 10 µM 5,5′,6,6′-tetrachloro-1,1′3,3′-tetraethyl-imidacarbocyanine iodide (JC-1) and incubated for 40 min at room temperature. The fluorescence intensity ratio of red (540 nm excitation and 590 nm emission) to green (490 nm excitation and 540 nm emission) was measured using a multi-well fluorescence reader [43].

### 2.11. Intracellular ATP Quantification

Intracellular ATP (ATP_i_) was measured using an ATP bioluminescent somatic cell test kit (Sigma-Aldrich). In brief, epimastigotes (1 × 10^7^ parasites per tube) were incubated in a solution containing 100 mM sucrose, 50 mM KCl, and 50 mM Tris-HCl in 0.1 mL (pH 7.2 adjusted with HCl). Cellular extracts were prepared by combining them with 0.1 mL of somatic cell ATP-releasing reagent and then chilling the mixture for 1 min. The mixture was transferred to MTS-11C mini tubes containing 0.1 mL ATP assay mix (*v*:*v*; Axygen) and swirled for 10 s at room temperature. A GloMax Multi JR detection system (Promega) measured the overall quantity of light emitted. In each experiment, the total intracellular ATP concentration per 10^7^ cells was determined using a standard ATP curve [29].

### 2.12. Glyceraldehyde-3-Phosphate Dehydrogenase (GAPDH) Activity

The reduction of NAD^+^ to NADH was used to assess the conversion of glyceraldehyde-3-phosphate to 1,3-bisphosphoglicerate, as described before [44], with slight modifications. Total epimastigote lysates were incubated for 15 min at 37 °C in reaction media containing 100 mM Triethalonamine:HCl buffer (pH 7.5) containing 1.0 mM EDTA, 5 mM MgSO_4_, 1.0 mM dithiothreitol, 1.5 mM NAD^+^, and 30 mM KH_2_AsO_4_ [45,46]. The reaction was started using 2 mM glyceraldehyde-3-phosphate, and the absorbance at 340 nm was measured every 1 min for 5 min. Total NADH generation was determined using the NADH standard curve.

### 2.13. Glucokinase Activity

Cellular extracts of epimastigotes were incubated in a reaction buffer containing 20 mM Tris-HCl (pH 7.4), 5 mM MgCl_2_, 1 mM glucose, 1 unit/mL glucose-6-phosphate dehydrogenase (G6PDH) (*Leuconostoc mesenteroides*), 0.1% Triton X-100, 1 mM NaF, 5 mM NaN_3_, 1 mM ATP, and 50–100 μg/mL protein [47]. After 3 min of incubation, the reactions were started by the addition of 0.5 mM β-NADP^+^, and quantified spectrophotometrically following the reduction of β-NADP^+^ to β-NADPH (λ = 340 nm) for 10 min. Total NADPH generation was determined using the NADPH standard curve.

### 2.14. Western Blotting

For western blotting detection, epimastigotes were lysed with 1 mL RIPA buffer supplemented with 1 mM phenylmethanesulfonyl fluoride and 5 mM leupeptin, for 30 min at 4 °C. Then, homogenates were centrifuged at 16,000 rpm for 10 min. Aliquots of the supernatants (containing 100 µg total protein) were separated by 12% SDS-PAGE and transferred to nitrocellulose membranes (Merck Millipore, Burlington, MA, USA), which were blocked with 5% milk in PBS plus 0.1% (*w*/*v*) Tween 20, probed overnight at 4 °C with the primary mouse anti-GAPDH antibody (1:500, Sigma-Aldrich), and detected using horseradish peroxidase (HRP)-conjugated anti-mouse IgG secondary antibody (1:5000, Santa Cruz Biotechnology, Dallas, TX, USA). The loading control was probed with a primary rabbit anti-tubulin antibody (1:500, Sigma-Aldrich) and detected using an HRP-conjugated anti-rabbit IgG secondary antibody (1:10,000, Santa Cruz Biotechnology). Luminescence was detected using an ImageQuant LAS 4000 digital imaging system (GE Healthcare Life Sciences, Amersham, UK) after the reaction with LuminataTM Forte Western HRP Substrate (Millipore, Billerica, MA, USA). Densitometric analysis was performed using ImageJ software version 1.50i (NIH Image, Bethesda, MD, USA) with background correction.

### 2.15. Superoxide Dismutase (SOD) Activity

Total SOD activity was assessed as previously described [29], based on SOD inhibiting the reduction of nitro blue tetrazolium (NBT) by O_2_^•−^. Epimastigote cells were harvested by centrifugation, washed three times in cold PBS, and disrupted by freeze-thaw. Centrifugation as described above was used to collect epimastigote cells, which were washed three times in cold PBS and disrupted by freeze-thaw. The Bradford method [42] was used to determine the protein content in the whole homogenate. In a final volume of 200 µL, the homogenates (using known amounts of protein in the range of 10–50 µg) were incubated in a reaction medium containing 45 mM potassium phosphate buffer (pH 7.8), 6.5 mM EDTA, and 50 mM NBT. The reaction began with the addition of 2 mM riboflavin. The sample’s absorbance at 560 nm was measured after 15 min in a lightbox. SOD activity was expressed as the quantity of enzyme-blocking NBT reduction by 50% for each amount of protein.

### 2.16. Amplex^®^ Red Peroxidase Assay

The rate of H_2_O_2_ reduction was assayed by the production of H_2_O_2_ in epimastigotes to H_2_O, which is stoichiometrically coupled (1:1) to the simultaneous oxidation of the non-fluorescent Amplex^®^ Red probe to the fluorescent resorufin [48]. Briefly, 10^7^ parasites/mL were incubated with 5 mM Tris-HCl (pH 7.4), 1.7 μM Amplex^®^ Red (Invitrogen, Carlsbad, CA, USA), and 6.7 U/mL horseradish peroxidase (Sigma-Aldrich) in a final volume of 100 µL, for 30 min at room temperature. Fluorescence evolution was observed at excitation/emission wavelengths of 563/587 nm. H_2_O_2_ concentration was determined using a standard curve.

### 2.17. Statistical Analysis

The data are provided as mean ± standard error of the mean (SEM). The unpaired Student’s *t*-test was used to compare two means. When comparing more than two means, a one-way ANOVA with Tukey’s test was applied, as specified in the text or the figure captions. The normal distribution was assessed before each ANOVA analysis. When results were expressed as a percentage of the RM group, the SEM was obtained from the absolute values. A comparison of the absolute results was carried out using the parametric test. Significance was set at *p* < 0.05. Except when otherwise indicated, different lowercase letters in superscripts indicate statistical differences among means in the same line of tables. Asterisks (also indicating *p* < 0.05) were used in figures and for comparing values from different lines in figures. GraphPad Prism 7.0 was used for statistical analysis and the preparation of figures (GraphPad Software, San Diego, CA, USA).

## 3. Results

### 3.1. Trypanosoma cruzi Epimastigotes Depend on Exogenous Fe Content for Fe Proliferation

The proliferation of *T. cruzi* epimastigotes during the exponential growth phase (Figure 1) depends on Fe in a concentration-dependent manner. In Fe-depleted medium (IDM, black circles), the maximum number of protozoa after 7 days decreased from 4.4 × 10^7^ to 1.0 × 10^7^ parasites/mL, compared to epimastigotes maintained in regular medium (RM, empty circles). The proliferative capacity of the cells was recovered with 8 µM Fe citrate added to the Fe-depleted medium (IDM + Fe, gray circles): the number of protozoa after 7 days increased from 1.0 × 10^7^ to 5.1 × 10^7^ parasites/mL, a condition similar to the control. In addition, Fe citrate supplementation was enough to restore intracellular Fe concentration compared to RM, while cells maintained at IDM presented low intracellular Fe content (Table 2). Although in IDM and in IDM + Fe there was an increase in the expression of Fe reductase (TcFR), the expression of the Fe transporter (TcIT) in cells maintained in IDM + Fe remained at levels similar to the RM condition (Table 2).

### 3.2. Exogenous Fe Selectively Modulates the Endocytic Capacity of Epimastigote Cells

Figure 2 shows that ionic Fe modulates both the function and structure of the endocytic pathway in *T. cruzi* epimastigotes. Figure 2A (see also representative cytometric analyses in Appendix A) demonstrates that parasites from IDM and IDM + Fe upregulate transferrin (Tf) uptake compared to control parasites grown in RM. However, hemoglobin (HB) uptake is upregulated only in IDM, compared to RM and IDM + Fe, indicating that free Fe in the culture medium controls hemoglobin uptake, in contrast with that observed with the transferrin uptake mechanism. Fe depletion stimulates bovine serum albumin (BSA) uptake in parasites grown in IDM, and the stimulation persists in IDM + Fe, indicating that this endocytic event is higher in both conditions. The fluorescence images in Figure 2B visually represent what we found in the cell cytometry data. Parasites upregulate the uptake of the endocytic tracers in IDM and IDM + Fe conditions.

Ultrastructural analysis of epimastigotes grown in the three different culture media revealed opposite trends depending on the organelles analyzed (Figure 3). While the general morphological aspect remained unmodified in IDM and IDM + Fe when compared with the parasites maintained in RM, important alterations can be seen in the reservosomes, lysosome-like organelles that are also involved in the storage of exogenous lipids, especially cholesterol, and enzymes involved in lipid synthesis from acetyl-CoA [49,50,51,52]. The homogeneous content of the reservosomes encountered in RM epimastigotes (upper line) contrasts with that seen in the other two groups. The depletion of Fe (middle line) provoked a marked increase in lipid accumulation, which was not reversed by the supplementation with Fe citrate (bottom line), indicating significant metabolic modifications.

### 3.3. Free Iron Supplementation-Induced Modifications in the Heme-Regulated Eukaryotic Initiation Factor 2α Kinase-Signaling Pathway

Facing the epimastigotes proliferation results presented in Figure 1 and considering that Fe depletion downregulates cytosolic and mitochondrial ribosomal proteins [53] via the heme-regulated inhibitor (HRI) of the translation through phosphorylation of the eukaryotic initiation factor 2α (eIF2α), we investigated two key elements of this signaling pathway (Table 3). Removal of heme and free Fe (IDM) increased the expression of HRI by approximately 90% and decreased by 50% the expression of eIF2α. In the case of HRI, the supplementation with free Fe restored the levels of HRI and upregulated those of eIF2α.

The faster response mediated by the HRI→eIF2α is integrated with the PKA pathway in several eukaryotic organisms [54], and the PKA pathway participates in the sensing of and response to nutrients, including Fe [55]. Table 3 also shows a decrease in PKA expression in epimastigotes grown in IDM and upregulation in IDM + Fe when both conditions are compared to RM. The impact of Fe depletion/supplementation was better seen with PKA activity: a pronounced decrease in parasites from IDM and an upregulation of more than 100% in IDM + Fe conditions (Table 3).

### 3.4. Exogenous Fe Alters Mitochondrial Respiratory Rates in T. cruzi Epimastigotes

As mentioned above, mitochondrial proteins decrease when the HRI→eIF2α pathway is dysfunctional [53], as encountered in the case of Fe depletion (Table 3). Fe concentration in the culture altered mitochondrial O_2_ consumption by *T. cruzi,* as observed in Table 4. The digitonin permeabilization of parasites did not alter the basal respiration profile. There was a difference in the O_2_ consumption after adding succinate (the consumption of O_2_ in the LEAK state) to IDM compared to RM and IDM + Fe (line 1). The O_2_ consumption in the presence of succinate without further additions matched the succinate-cytochrome c oxidoreductase activity (line 5). However, after additions of ADP (oxidative phosphorylation state-OXPHOS) (line 2) and H^+^ ionophore FCCP (electron transfer system uncoupled from phosphorylation-ETS) (line 3), there was a significant and similar decrease in O_2_ consumption in both IDM and IDM + Fe. After adding antimycin A to inhibit the mitochondrial complex III, the residual respiration (ROX) was almost completely abolished under all conditions (line 4). The impact of Fe depletion (even after Fe citrate supplementation) on mitochondrial internal membrane potential (ΔΨ_m_) can be seen in line 6. The ratio between the safranin A fluorescence in the absence and presence of FCCP (F/F_FCCP_) increased by 600% in parasites grown in IDM and attained an intermediary level ~200% higher in IDM + Fe conditions.

Parasites maintained in the RM medium presented intracellular ATP content (ATP_i_) more than twice as high when compared to parasites grown in IDM or IDM + Fe media (Table 4, line 7). The addition of oligomycin to block the F_o_F_1_-ATP synthase [56] decreased ATP_i_ content by more than 50% with respect to the RM group without inhibitors (line 8), indicating that, under these conditions, the cellular ATP supply comes from mitochondrial activity. Moreover, the addition of iodoacetamide (line 9), an inhibitor of glyceraldehyde 3-phosphate dehydrogenase (GAPDH): (i) reduced ATP_i_ in parasites grown in RM to levels that are similar to those encountered in IDM and IDM + Fe conditions in the absence or presence of oligomycin; (ii) provoked a further 50% decrease in ATP_i_ from IDM and IDM + Fe parasites when compared to that grown in RM, indicating that the glycolytic pathway is responsible for ATP production in IDM and IDM + Fe conditions.

### 3.5. Upregulation of Glycolytic Enzymes in Parasites Grown in Fe-Depleted Media

The activity and abundance of two glycolytic enzymes were evaluated in the experiments depicted in Figure 4. Epimastigotes grown in IDM and IDM + Fe presented, respectively, with GAPDH activities two and three times higher than those grown in RM (Figure 4A), an increase that matches the higher expression of the enzyme (Figure 4B). The abundance of the enzyme was also investigated; the levels increased by more than 100% in IDM + Fe parasites compared with the RM group without modification in the IDM group (Figure 4C). The shift toward a glycolytic metabolic profile was confirmed by the more than 200% increase in the activity of glucokinase (Figure 4D), the enzyme that connects the pentose phosphate pathway and the glycolytic pathway, which are the two major pathways for glucose metabolism in *T. cruzi* [57].

### 3.6. Modifications in Redox Signaling Induced by Depletion of Fe in Culture Medium

The mutual regulation of hexokinases and redox signaling comprises important events in parasites that cause severe diseases, e.g., *T. cruzi* [58]. This is the reason why we decided to investigate two enzymes involved in *T. cruzi* redox signaling and see the influence of Fe depletion: (i) superoxide dismutase (TcSOD), which has Fe as a cofactor in *T. cruzi* [59], and (ii) ascorbate peroxidase (TcAPX) [60], an enzyme that is absent in the human host [61], which metabolizes H_2_O_2_ to H_2_O. We measured TcSOD activity by following two strategies. First, by evaluating the formation of nitroblue tetrazolium formazan (NBT formazan) after oxidation of xanthine to uric acid and generation of O_2_^•−^, a reaction in which the higher superoxide dismutase activity is, the lower NBT formazan formation will be. Second, measuring the H_2_O_2 _formed through different processes, including the premature transfer of electrons to O_2_ in the *T. cruzi* mitochondrion. Table 5 shows that the SOD activity associated with the O_2_^•−^ formed during the oxidation of xanthine increased by approximately 200% in parasites grown in IDM regardless of supplementation with Fe. Moreover, when the total H_2_O_2_ production was assayed, a significant decrease of about 40% was found in IDM parasites, which was suppressed in the IDM + Fe group. Finally, the expression profile of TcSOD was the mirror of the activity, whereas that of TcAPX matched the activity, i.e., decreased in the parasites grown in IDM, with recovered levels in the IDM + Fe group.

### 3.7. Differentiation of Epimastigotes Is Lower in Fe-Depleted Medium and Is Recovered after Fe Supplementation

Differentiation is a crucial step in the life cycle of *T. cruzi* because it ensures the success of infection and because the redox status of the parasitic environment is an important modulator [62]. Figure 5 demonstrates that, in culture, differentiation is critically dependent on the presence of Fe in the medium, and this dependence is more evident at initial times. While differentiation was barely detectable along the first day in IDM (filled circles), it exhibited a linear behavior in RM (empty circles) and depicted a burst in IDM + Fe (gray circles). Remarkably, the initial differentiation rate slop doubled in this group regarding RM parasites, but it seemed to tend toward a lower plateau from the 3rd day of culture.

## 4. Discussion

The central findings in the present study reveal a key role of medium ionic Fe in the proliferation of *T. cruzi* epimastigotes, with Fe depletion promoting increased oxidative stress, selective modifications in the intracellular ATP content, alterations in the HRI→eIF2α and PKA signaling pathways, increased lipid accumulation in the reservosomes, decreased mitochondrial function, and inhibition of differentiation toward trypomastigotes, in a metabolic condition shifted from respiration to glycolysis. This ensemble of results points to a pleiotropic function of ionic Fe in connected processes and pathways in *T. cruzi* epimastigotes. Using the Dm28c strain, which differentiates from trypomastigotes, allowed us to investigate the influence of exogenous Fe on the evolution of epimastigotes to trypomastigotes and, therefore, on a vital step of the parasite’s life cycle.

The Fe depletion-induced lower intracellular content of Fe (Table 2). Notably, in the IDM + Fe medium, the expression of TcFR increased without a parallel increase in TcIT transcription. Even though TcFR and TcIT are coupled in the process of Fe uptake by the parasite, their expression is differentially modulated by the intracellular Fe content. Free Fe could regulate the TcIT transcription, so in the IDM + Fe medium, TcIT is downregulated relative to the TcIT transcript in IDM. Differently, TcFR uses Fe-containing proteins for the reaction Fe^3+^→Fe^2+^, and since these proteins (hemin and transferrin) were neither added to IDM nor to IDM + Fe, TcFR is upregulated in both cases (Table 2).

The decreased succinate-cytochrome c oxidoreductase activity, the dropped O_2_ consumption in the presence of normal partial pressure of O_2_, and the lower intracellular ATP (Table 4) are indicative of mitochondrial damage, as proposed several years ago [63]. The impairment of the mitochondrial function seems to be functional because the ultrastructure of the organelle is preserved (Figure 3). Therefore, it could be hypothesized that Fe starvation promotes dysfunction at the level of the iron-sulfur clusters in the heterodimeric SDH2_N_:SDH2_C_ subunit described in the mitochondrial complex II from *T. cruzi* [64], as well as in the FoF_1-_ATPase [65], a possibility that emerges from the accentuated inhibition of respiration in the presence of ADP (the oxphos state) and the increased ΔΨ_m_ (Table 4). Although succinate-cytochrome c oxidoreductase activity is restored by Fe-citrate supplementation (Table 4), mitochondrial function impairment seems linked to Fe-protein depletion, especially hemin. It has been demonstrated that, in epimastigotes, heme changes mitochondrial physiology [40]. NADH-ubiquinone oxidoreductase gene (0.8-fold) and succinate dehydrogenase (1.40-fold) are upregulated in the presence of heme. Besides, heme influences *T. cruzi* epimastigote energy metabolism. The contribution to ATP synthesis may depend on glycosomal fermentation, which provides energy support for the parasite’s growth, an establishment inside the vector [66], and differentiation into trypomastigotes (Figure 5).

The proposal that Fe and hemin depletion promotes a shift from oxidative metabolism to a glycolytic one is reinforced by the increased GADPH expression and activity (Figure 4A,B) and the increased glucokinase activity (Figure 4D). The upregulated glycolytic and pentose phosphate pathways, which are considered central for the glucose metabolism in *T. cruzi* [57], likely provide acetyl-CoA and NADPH, respectively, for the proposed increase of fatty acid synthesis accumulation within the reservosomes (Figure 3). As mentioned earlier, these organelles present a varied repertoire of enzymes that catalyze different lipid metabolism pathways [50], and for this reason, lipid accumulation in the reservosomes deserves special discussion in the context of the other enzyme modifications and proliferation.

The impairment of the IDM in the life cycle of epimastigotes could be linked to the alterations encountered in HRI→eIF2α and PKA signaling. The increased HRI, which phosphorylates eIF2α [54], together with the downregulation of eIF2α itself (Table 3), likely culminate in repressed gene expression and overall protein translation, thus compromising the evolution of the parasite in the Fe-deprived medium. The pronounced downregulation of PKA activity (and expression) could be associated with the downregulation of lipase activity and fatty acid release and oxidation. It may be that decreased PKA activity in IDM parasites (Table 3) results in the inhibition of a PKA-modulated lipase and functional immobilization of the lipids in the reservosomes. This proposal is supported by the fact that PKA recovery and expression increased to levels even higher than in RM conditions after Fe supplementation (Table 3), which ensures lipid turnover and recovery of parasite evolution as discussed above. As additional support for this view, it is noteworthy that the genetic inhibition of PKA is lethal for *T. cruzi* [67].

Lipolysis in *T. cruzi* is associated with glucose metabolism [57]. For this reason, the upregulation of central enzymes of the glycolytic pathway, GADPH, and glucokinase, in both IDM and IDM + Fe media (Figure 4), leads us to hypothesize that the absence of hemin is central to the upregulation of glycolysis and the pentose phosphate pathway, but that replenishing of ionic Fe is responsible for the possible stimulation of lipid hydrolysis, glycerol release, formation of glycerol 3-phosphate catalyzed by a Tc-glycerol kinase [68], and further feeding of the glycolytic pathway. The other metabolic branch after Fe-stimulated lipid turnover, the β-oxidation of fatty acids [69], can feed the acetyl-CoA pool in the epimastigotes cell, further stimulating the formation of ATP via its condensation with succinate, synthesis of succinyl-CoA, and recycling of succinate with the release of CoA, as proposed in genomic studies carried out in *T. cruzi* [57] and earlier demonstrated in *T. brucei* [70].

Ionic Fe and heme depletion lead to a downregulation of total FeSOD, regardless of FeSOD origin. It is possible that SOD activity is higher in epimastigotes maintained at Fe/heme or heme depletion, demonstrating a compensating mechanism, probably due to higher activity of cytosolic and mitochondrial FeSOD (FeSODB and FeSODA). While FeSODB has a crucial role in the defense of parasites against O_2_^•−^ [71], FeSODA is related to mitochondrial redox balance and generates the signaling molecule for amastigote differentiation, H_2_O_2_ [72]. A downregulation of H_2_O_2_ levels in Fe depletion conditions probably deregulates parasite differentiation. These low H_2_O_2_ levels could be due to a non-enzymatic system besides the glutathione ascorbate cycle. Recently, it was demonstrated that *T. cruzi* trypomastigotes employ ROS as a signaling molecule to differentiate, whereas epimastigotes use ROS to proliferate rather than differentiate [72].

Finally, Figure 6 presents a hypothetical mechanistic model regarding the overall mechanisms occurring during Fe depletion or Fe supplementation. Although the metabolic shift occurs in both cases, the ROS formation and pathway signaling present slight differences that culminate in differentiation/proliferation impairment (Figure 6A), which is restored by Fe supplementation (Figure 6B). In conclusion, although heme (or Fe-containing proteins) is essential for a functional mitochondrial metabolism, exogenous Fe is required for proper signaling to control parasite proliferation and H_2_O_2_ formation, which stimulate parasite differentiation, thus interfering with parasite virulence. These related mechanisms and processes modulated by exogenous ionic Fe have implications for human health because, by providing energy support for parasite growth and differentiation, they ensure the continuity of the *T. cruzi* life cycle and the propagation of Chagas disease.

## Figures and Tables

**Figure 1 antioxidants-12-00984-f001:**
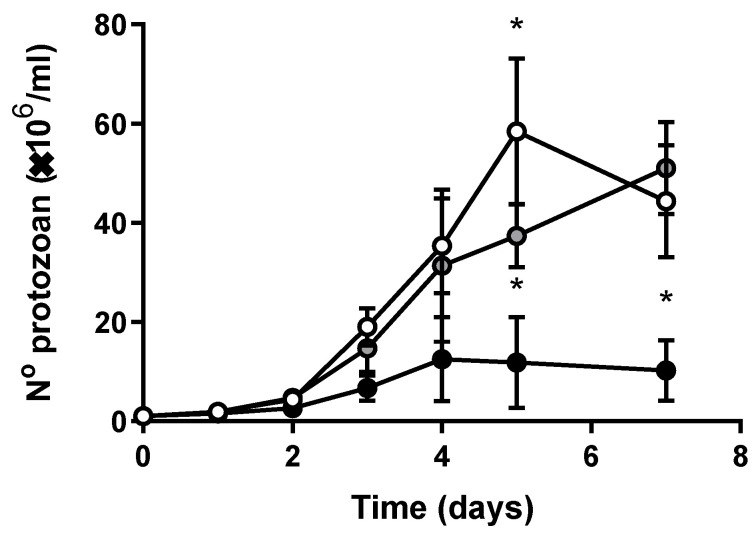
The effect of exogenous Fe on *T. cruzi* growth. *Trypanosoma cruzi* epimastigotes were harvested, washed twice, seeded in fresh medium, and grown for the indicated times in Regular Media (RM: Brain Heart Infusion medium (BHI) supplemented with 30 µM hemin and 10% Fetal Bovine Serum (FBS)) (white circles), Iron-Depleted Media (IDM: BHI without hemin supplementation and treated with Chelex for Fe depletion, supplied with 10% Iron-depleted FBS) (black circles), or in Iron-Depleted Media supplemented with Fe-citrate (IDM + Fe: IDM described before, plus 8 µM Fe-citrate) (gray circles), (n = 6). Using one-way ANOVA with Tukey’s test, we assessed differences between mean values at time-matched determinations; * *p* < 0.05 with respect to RM.

**Figure 2 antioxidants-12-00984-f002:**
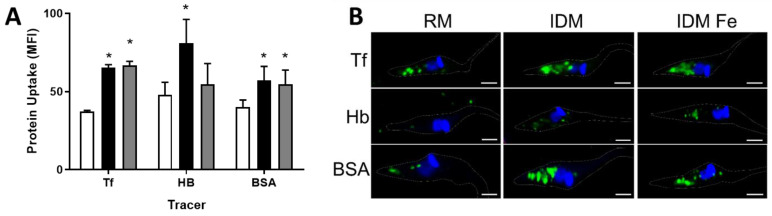
Fe exogenous content alters endocytosis in epimastigotes. Epimastigotes were incubated in RPMI medium containing transferrin–FITC (Tf), hemoglobin–FITC (HB), or BSA–FITC (BSA) for 30 min at 28 °C. (**A**) Endocytosis assay evaluated by flow cytometry in epimastigotes maintained at RM (white bars), IDM (black bars), or IDM + Fe (gray bars) (n = 4 from different cultures); * *p* < 0.05; (**B**) immunofluorescence of epimastigotes at different culture conditions and endocytic tracers incubation. Endocytic content staining with fluorescence macromolecules transferrin–FITC (Tf), hemoglobin–FITC (HB), or BSA–FITC (BSA) (green). Nuclei staining with DAPI (blue). Bars: 10 µm.

**Figure 3 antioxidants-12-00984-f003:**
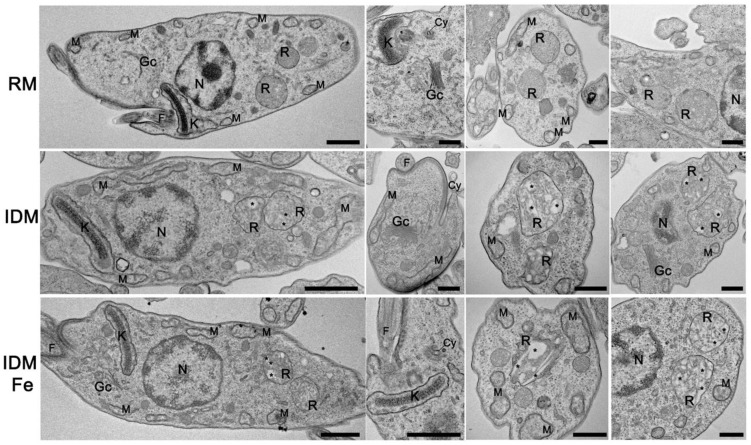
Ultrastructural changes in *T. cruzi* parasites submitted to different exogenous Fe contents. Epimastigotes were processed and observed by transmission electron microscopy. (Upper line) Epimastigotes maintained in RM show normal morphology of major cellular organelles such as the kinetoplast (K), nucleus (N), golgi complex (Gc), cytostome-cytopharynx complex (Cy), reservosomes (R), mitochondria (M), and flagellum (F). (Middle line) Epimastigotes were maintained in IDM. (Bottom line) Epimastigotes maintained in IDM + Fe. Epimastigotes maintained at IDM or IDM + Fe media present typical organelles morphology and positioning compared to those maintained at RM, except for reservosomes, which present a higher content of lipid inclusions (asterisks) compared to the ones in RM. Bars: 500 nm. Five culture samples were examined.

**Figure 4 antioxidants-12-00984-f004:**
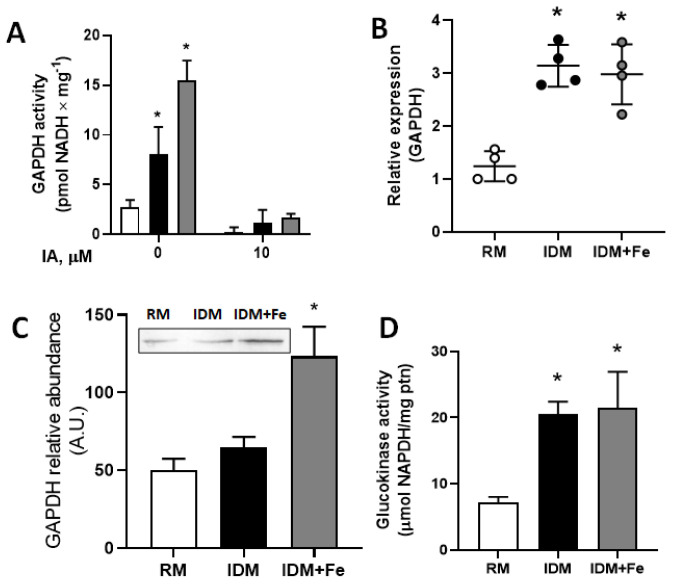
Exogenous Fe leads to glycolytic changes in epimastigotes. (**A**) GAPDH activity in the absence or presence of iodoacetamide (IAA, 10 µM), as indicated on the abscissa, in epimastigotes in a mid-log phase when maintained in RM (empty bar), IDM (black bar), or IDM + Fe (gray bar). The absorbance due to the formation of NADH was monitored at 340 nm (*n* = 4); (**B**) quantification of the GAPDH transcript in *T. cruzi* epimastigotes. Quantitative PCR was done using 100 ng cDNA from epimastigotes maintained at RM (empty circles), IDM (black circles), or IDM + Fe (gray circles), (*n* = 4); (**C**) densitometric analysis of Western blot results (*n* = 4), with the inset showing representative Western blotting analysis. Membranes were probed with primary mouse anti-GAPDH antibody (1:500) as described in the Materials and Methods section; (**D**) glucokinase activity was measured as described in Materials and Methods in epimastigotes maintained at RM (empty bar), IDM (black bar), or IDM + Fe (gray bar). The absorbance due to the formation of NADPH was monitored at 340 nm (*n* = 4); * *p* < 0.05 in all cases with respect to RM. We used a one-way ANOVA with Tukey’s test to assess differences between mean values.

**Figure 5 antioxidants-12-00984-f005:**
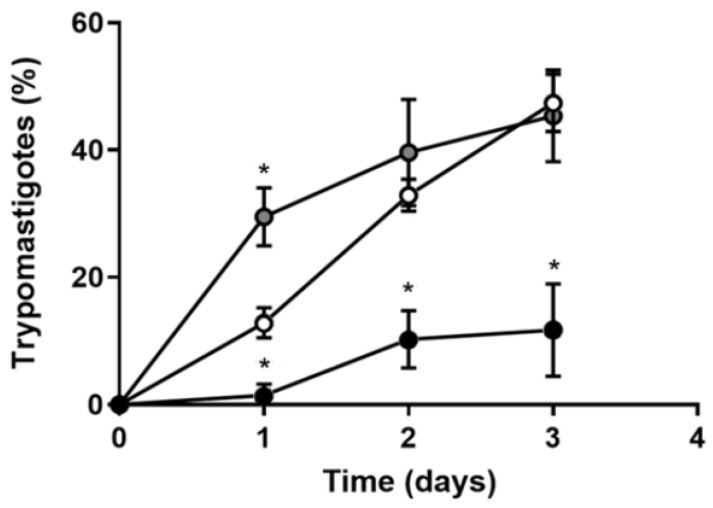
Fe depletion arrests *T. cruzi* differentiation to trypomastigotes. Epimastigotes cultivated in RM (empty circles), IDM (black circles), or IDM + Fe (gray circles) were submitted to in vitro differentiation media TAU. Each day, epimastigotes and trypomastigotes were differentially counted to determine the percentage of trypomastigotes in culture (*n* = 4); * *p* < 0.05 comparing time-matched determinations with respect to RM. Using one-way ANOVA with Tukey’s test, we assessed differences between mean values.

**Figure 6 antioxidants-12-00984-f006:**
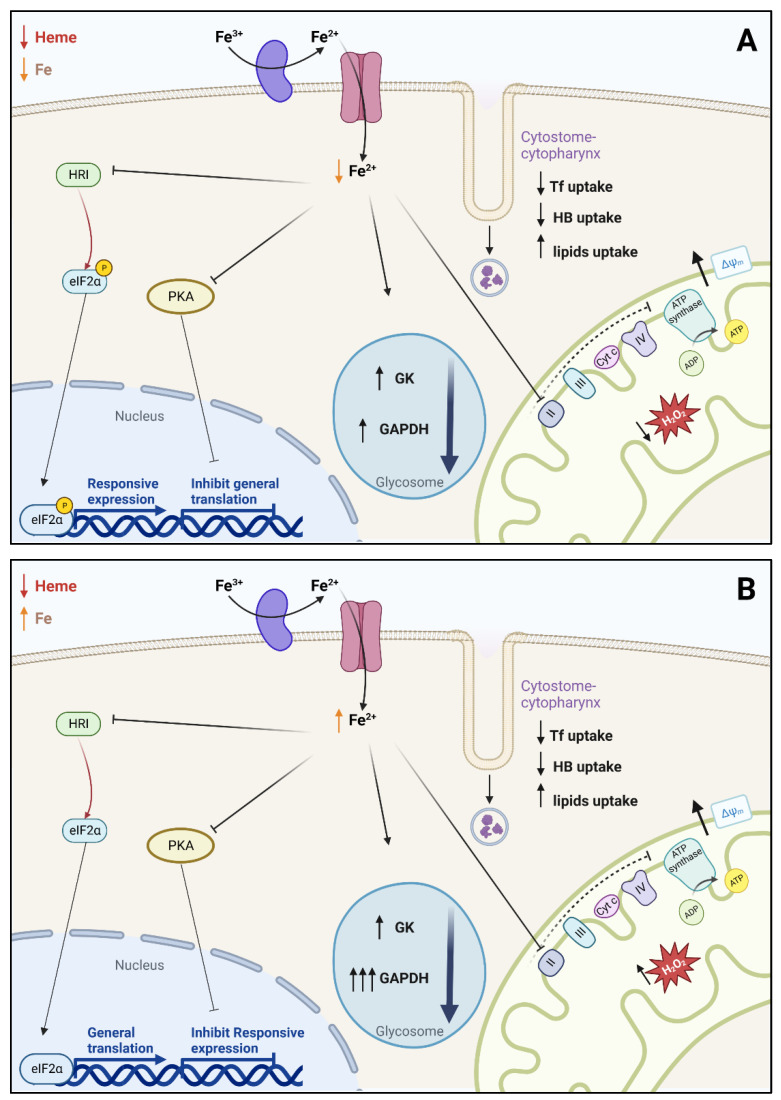
A proposed model describing the main events occurring on *T. cruzi* submitted to (**A**) Fe depletion or (**B**) Fe supplementation in low heme concentration (left outside part of the figure), which culminate in decreased or restored intracellular Fe^2+^, as demonstrated by the opposite direction of the orange arrow near the upper part of the figure. The decrease in intracellular Fe in (**A**) was 50% in parasites grown in IDM (Table 2). The vesicles near the cell membrane (cytostome-cytopharynx region) indicate the selective modifications in the endocytic protein and lipid uptake. The left part of the arrow ensembles represents the modifications of the HRI→eIF2α and PKA signaling pathways and the different modifications in expression and translation. The blue circles in the middle of the panels present the quantitative differences in GAPDH abundance and activity. The blunt black arrows (in the right part of both panels) point to the impaired mitochondrial function provoked by Fe depletion, which is not sufficiently restored by Fe supplementation. The red stars in the lower right corner depict the opposite effects of Fe depletion and Fe supplementation on H_2_O_2_ formation. The figure was designed using BioRender.com.

**Table 1 antioxidants-12-00984-t001:** Primer sequences for nine genes analyzed.

Primers Name	Forward Primer	Reverse Primer
*T. cruzi* iron transporter	TCTGGTCGCTTCTCTTCTCG	TAAAGACTCCGGCACACAGT
*T. cruzi* ferric reductase	GTGGTTTGTAGACCGGCTGT	GTGCCATTGCAAGAGAGACA
*T. cruzi* heme-regulated inhibitor	CATTGTGGAGGCGTTGGAAA	TGGAAGAGCACCGTGAAGAT
*T. cruzi* eukaryotic initiation factor 2α	CCGTTTAACGTTCCCTTTGA	GTCCCAGCTCGTTACTCCAA
*T. cruzi* protein kinase A	CCGGGTGTACTTTGTGTTGG	CGCAAACCCAAAGTCAGTCA
*T. cruzi* glyceraldehyde-3-phosphate dehydrogenase	GCAAGCTTGGTGTGGAGTAC	CTCACTGGGGTTGTACTCGT
*T. cruzi* superoxide dismutase	GTCGGATATTGTGTTGGGCC	CCCTGTACCACGGAAACTCT
*T. cruzi* ascorbate peroxidase	CACGACAAGTACGGCTTTGA	CTTCGCGATGGAACGATATT
*T. cruzi* tubulin	AAGCGCACGATTCAGTTTGT	CTCCATACCCTCACCAACGT

**Table 2 antioxidants-12-00984-t002:** Intracellular Fe content and quantification of the TcFR and TcIT transcripts *.

	RM	IDM	IDM + Fe
Fe content	22.07 ± 0.78 ^a^	11.01 ± 2.42 ^b^	26.59 ± 3.31 ^a^
TcFR transcript	0.96 ± 0.05 ^a^	1.59 ± 0.11 ^b^	2.14 ± 0.16 ^b^
TcIT transcript	0.85 ± 0.05 ^a^	1.72 ± 0.24 ^b^	0.72 ± 0.12 ^a^

** Trypanosoma cruzi* epimastigotes were maintained in Regular Media (RM), Iron-Depleted Media (IDM), or Iron-Depleted Media plus 8 µM Fe-citrate (IDM + Fe). Intracellular Fe content determination (*n* = 5) and quantitative PCR for TcFR (*n* = 5) or TcIT (*n* = 5) (using 100 ng cDNA) were carried out in epimastigotes in a mid-log phase when maintained in the different media. Using one-way ANOVA with Tukey’s test, we assessed differences between mean values. Different lower-case letters as superscripts in the same line indicate different mean values (*p* < 0.05).

**Table 3 antioxidants-12-00984-t003:** Influence of PKA pathway on the response to exogenous ionic Fe in *T. cruzi* epimastigotes *.

	RM	IDM	IDM + Fe
TcHRI transcript	1.04 ± 0.02 ^a^	1.70 ± 0.11 ^b^	1.44 ± 0.25 ^a^
eIF2α transcript	1.00 ± 0.07 ^a^	0.41 ± 0.05 ^b^	1.45 ± 0.05 ^b^
TcPKA transcript	1.04 ± 0.04 ^a^	0.78 ± 0.07 ^a^	1.57 ± 0.16 ^b^
PKA activity	0.98 ± 0.02 ^a^	0.29 ± 0.08 ^b^	2.12 ± 0.24 ^a^

* Quantification of the TcHRI (*n* = 5), elF2α (*n* = 5), and TcPKA (*n* = 5) transcripts in *T. cruzi* epimastigotes. Quantitative PCR was done using 100 ng of cDNA from epimastigotes in a mid-log phase when maintained in RM, IDM, or IDM + Fe. PKA activity was measured, as described in Materials and Methods (*n* = 5), in epimastigotes maintained in RM, IDM, or IDM + Fe. In all cases, differences were assessed using one-way ANOVA with Tukey’s test. Different lower-case letters as superscripts in the same line indicate different mean values.

**Table 4 antioxidants-12-00984-t004:** Removal of exogenous Fe leads to mitochondrial alterations in epimastigotes *.

Line	Determination	Additions	RM	IDM	IDM + Fe
	Mitochondrial respiration state				
1	LEAK	Succinate	0.87± 0.13 ^a^	0.38 ± 0.06 ^b^	0.66 ± 0.09 ^a^
2	OSPHOS	ADP	1.51 ± 0.16 ^a^	0.44 ± 0.07 ^b^	0.73 ± 0.09 ^b^
3	ETS	FCCP	2.16 ± 0.26 ^a^	0.54 ± 0.10 ^b^	0.92 ± 0.10 ^b^
4	ROX	Antimycin A	0.33 ± 0.02	0.23 ± 0.01	0.12 ± 0.02
5	Succinate cytochrome c oxidoreductase activity	Succinate	103.5 ± 5.7 ^a^	41.1 ± 7.9 ^b^	93.9 ± 14.4 ^a^
6	ΔΨ_m_ (F/F_FCCP_)	Succinate + FCCP	1.35 ± 0.03 ^a^	9.23 ± 0.57 ^b^	2.96 ± 0.59 ^c^
7	ATP_i_	No inhibitors	24.24 ± 0.96 ^a^	14.28 ± 1.52 ^b^	12.51 ± 2.09 ^b^
8		Oligomycin	10.63 ± 0.57 ^a^	7.79 ± 0.94 ^a^	8.98 ± 2.25 ^a^
9		Iodoacetamide	11.74 ± 2.45 ^a^	6.42 ± 1.14 ^b^	6.28 ± 0.54 ^b^

* *Trypanosoma cruzi* epimastigotes maintained in RM, IDM, or IDM + Fe media were tested for respiration of intact epimastigotes. The mitochondrial respiration, the succinate cytochrome c oxidoreductase activity, and the ATP_i_ are expressed in pmol of O_2_ consumed per million of cells, % of the activity in RM, and nmol of intracellular ATP per 10^7^ cells, respectively. Epimastigotes were digitonin-permeabilized, as described in Nogueira et al. [40], to measure mitochondrial respiration after the successive additions of succinate, ADP, FCCP, and antimycin A (*n* = 3) (lines 1–4). Succinate-cytochrome c oxidoreductase activity was measured as the rate of ferricytochrome c reduction upon adding succinate in epimastigotes (*n* = 3) (line 5). The F/F_FCCP_ ratio was used to estimate the ΔΨ_m_, where F is the mean fluorescence intensity in the absence of the uncoupler FCCP and F/F_FCCP_ is the mean fluorescence in the presence of FCCP (*n* = 3) (line 6). Intracellular ATP in *T. cruzi* epimastigotes was measured in the absence or presence of oligomycin or iodoacetamide (*n* = 3) (lines 7–9). We assessed differences between mean values using a one-way ANOVA with Tukey’s test. Different lower-case letters as superscripts indicate statistical differences among the mean values in the same line (*p* < 0.05). Asterisks in lines 8 and 9 indicate a statistical difference (*p* < 0.05) with respect to RM in the absence of inhibitors.

**Table 5 antioxidants-12-00984-t005:** Effect of exogenous Fe on redox signaling *.

	RM	IDM	IDM + Fe
SOD activity	7.98 ± 2.26 ^a^	21.47 ± 3.08 ^b^	23.24 ± 3.94 ^b^
Production of H_2_O_2_	55.01 ± 0.75 ^a^	33.59 ± 6.33 ^b^	60.50 ± 5.34 ^a^
TcSOD transcript	1.01 ± 0.01 ^a^	0.45 ± 0.09 ^b^	0.44 ± 0.06 ^b^
TcAPX	1.17 ± 0.10 ^a^	0.14 ± 0.04 ^b^	1.29 ± 0.20 ^a^

* SOD activity (*n* = 4) and production of H_2_O_2_ (*n* = 3) were assessed in living epimastigotes maintained in RM, IDM, or IDM + Fe. The quantification of the TcSOD (*n* = 5) and of the TcAPX transcripts (*n* = 5) in *T. cruzi* epimastigotes was done using 100 ng cDNA from epimastigotes grown in each culture medium. We used one-way ANOVA with Tukey’s test to assess differences between mean values; Different lower-case letters as superscripts indicate statistical differences among the mean values in the same line (*p* < 0.05).

## Data Availability

The original contributions presented in the study are included in the article and the Appendix A. Further inquiries can be directed to the corresponding author.

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
