# Peer review of "Iron Uptake Controls Trypanosoma cruzi Metabolic Shift and Cell Proliferation"

_antioxidants, 2023, doi:10.3390/antiox12050984_

Round 1

Reviewer 1 Report

Dear Authors,

Thank you for the opportunity to review this paper. In my opinion, the article is substantively valuable and well written.

In my opinion, the publication requires minor corrections and additions.

Comments and Suggestions for Authors:

1. The words used in the title are also keywords, so there should be different words in the keywords than in the title. This increases the likelihood that readers will find your publication. Please correct.

2. Please complete the publication with explanations of all abbreviations used. 

3. Please describe in detail the statistical methods used and the programs used in which they were performed. I have doubts whether the assumptions of normality (and a number of others) for the applied analysis of variance have been met. Also, could you use parametric tests? What alpha levels are adopted? Please provide relevant details.

4. Despite the very interesting Figure, in my opinion, there is a lack of tabular summaries presenting specific numerical values of the results obtained. I believe that only in tables can we precisely present scientific results, and Figures are graphical supplements. This form, I think, will be conducive to citations. Please consider including the tables, partly in publications, and partly in supplementary materials.

Kind regards,

reviewer

Author Response

Reviewer 1.

General comment.

Citation: “Thank you for the opportunity to review this paper. In my opinion, the article is substantively valuable and well written.”

Answer: Thank you for your comment.

Specific comments.

Comment 1.

Citation: “The words used in the title are also keywords, so there should be different words in the keywords than in the title. This increases the likelihood that readers will find your publication. Please correct.”

Answer: We replaced 4 words in the keywords for other 4 that are different from those found in the title (page 2, lines 36 to 37; RM).

Comment 2.

Citation: “Please complete the publication with explanations of all abbreviations used.”

Answer: We added a list of abbreviations with their corresponding explanations in the RM (page 18, line 703 to page 19, line 747), as recommended.

Comment 3.

Citation: “Please describe in detail the statistical methods used and the programs used in which they were performed. I have doubts whether the assumptions of normality (and a number of others) for the applied analysis of variance have been met. Also, could you use parametric tests? What alpha levels are adopted? Please provide relevant details.”

Answer: The statistic section 2.17 in the RM now provides all the details requested by the Reviewer. Please compare this modified section in the RM (page 7, lines 305 to 315), with that presented in OM.

Comment 4.

Citation: “Despite the very interesting Figure, in my opinion, there is a lack of tabular summaries presenting specific numerical values of the results obtained. I believe that only in tables can we precisely present scientific results, and Figures are graphical supplements. This form, I think, will be conducive to citations. Please consider including the tables, partly in publications, and partly in supplementary materials.”

Answer: This comment is addressed as follows.

(i) Fig. 1 in OM. Panel 1A is the new Fig. 1 with a single panel in RM (page 8). It was preserved because it presents a time course of growing. Panels B–D became Table 2 in RM (page 8). Different lower-case letters as superscripts indicate different mean values in the same line.

(ii) Fig. 2 in OM is Fig. 2 in RM (page 9) because it presents fluorescence images.

(iii) Fig. 3 in OM is also Fig. 3 in RM (page 10) because it shows transmission electron microscopy images.

(iv) Fig. 4 in OM. The results from all panels are now presented in the new Table 3 (RM, page 11).

(v) Fig. 5 in OM. The results from all panels are now presented in the new Table 4 (RM, page 12).

(vi) Fig. 6 in OM. It is preserved as Fig. 4 in RM (page 13) because it presents a Western blotting image.

(vii) Fig. 7 in OM. The results from all panels are now presented in the new Table 5 (RM, page 14).

(viii) Fig. 8 in OM. It is preserved as Fig. 5 in RM (page 14) because it presents the time course of differentiation from epimastigotes to trypomastigotes.

(ix) Fig. 9 in OM. It is preserved as Fig. 6 in RM (page 17) because it presents a hypothetical mechanistic model for the consequences of exogenous Fe depletion.

Please, note that the legends of figures replaced by tables are now used as legends of the new tables. The changes are mirrored in the Results and Discussion sections of the RM when referring to the figures and tables (see highlights throughout these sections and in the figure legends in the marked version of RM).

Reviewer 2 Report

Current report investigated the transport of iron (Fe) in Chagas Disease etiologic parasite Trypanosoma cruzi at various cellular processes. I like to give the following comments.

1.      Findings did not show in abstract and novelty is hard to understand.

2.      Main aim of current study is important in the introduction section.

3.      In the last of introduction section, “Fe is a critical micronutrient for trypanosomatids, as previously stated” which needs the reference.

4.      In the methods, FeCl3 concentrations (0–75 μM; [37]) used to calculate the Fe content must show the supplier of FeCl3 in clear.

5.      In Table 1, Primer needs the full name in the legends.

6.      Sample size seems too small (N<5) in each group. Therefore, expression of the data shall be the same, particularly the group in column.

7.      Figure 9 showed a hypothetical mechanistic model of current findings. What is the contribution in human health? Please describe it in detail.

8.      In Figure 9, Fe depletion shown in A site seems hard to follow because intracellular Fe was raised. Please check it carefully.

Author Response

Reviewer 2.

Comment 1.

Citation: “Findings did not show in abstract and novelty is hard to understand.”

Answer: The abstract was totally rewritten in the RM (see comment 1 of Reviewer 4, below). The modified abstract in the RM now summarizes the findings in its section 3 (page 1, line 30 to page 2, line 33).

Comment 2.

Citation: “Main aim of current study is important in the introduction section.”

Answer: We present the main aim of the study in the modified introduction section (page 3, lines 101 to 104; RM).

Comment 3.     

Citation: “In the last of introduction section, “Fe is a critical micronutrient for trypanosomatids, as previously stated” which needs the reference.

Answer: Proper references (# 9–12) have been included after the sentence “Fe is a critical micronutrient for trypanosomatids, as previously stated” (page 3, lines 98 to 99); RM).

Comment 4.     

Citation: “In the methods, FeCl3 concentrations (0–75 μM; [37]) used to calculate the Fe content must show the supplier of FeCl3 in clear.”

Answer: The supplier of FeCl3 is provided in the RM (page 4, lines 150 to 151).

Comment 5.     

Citation: “In Table 1, Primer needs the full name in the legends.”

Answer: The modified Table 1 now shows the full name of the primers (page 4; RM).

Comment 6.

Citation: “Sample size seems too small (N<5) in each group. Therefore, expression of the data shall be the same, particularly the group in column.”

Answer: The RM now presents all the qPCR analysis with a greater number of experiments carried out with different parasite cultures. Please, see Tables 2, 3 and 5 on pages 8, 11 and 14 (RM).

Comment 7.     

Citation: “Figure 9 showed a hypothetical mechanistic model of current findings. What is the contribution in human health? Please describe it in detail.

Answer: The contribution in human health is described on page 16, lines 642 to 646 (RM) in the last sentence of the Discussion section.

Comment 8.     

Citation: “In Figure 9, Fe depletion shown in A site seems hard to follow because intracellular Fe was raised. Please check it carefully.”

Answer: We admit that the legend of Fig. 9 in OM (Fig. 6 in RM; see our answer (ix) to comment 4 by Reviewer 1) was unclear with respect to the decrease of intracellular Fe content in epimastigotes grown in IDM. The modified legend of this figure (page 17, lines 674 to 675; RM) now clearly mentions that intracellular Fe content decreased by 50% (Fig. 1B in OM; Table 2 in RM) in epimastigotes grown in iron-depleted medium (panel A). Please, see also our answer to comment 9 by Reviewer 3.

Reviewer 3 Report

The manuscript antioxidants-2320474 reports a comprehensive analysis on the effect of iron depletion and repletion in energetic metabolic pathways, intracellular ROS formation, and proliferation and differentiation of epimastigotes of Trypanosoma cruzi. The description of metabolic adaptations, among other results, is an important advance to improve our understanding of the parasite’ plasticity. Technically, the study is sound, and the different methodologies used for answering the questions are fair. Authors are encouraged to address some few issues to improve the manuscript.

Introduction page 1. “ However, several cases have lately been reported in Europe, North America, and Japan. As a result, Chagas disease became a significant public health concern worldwide [3]”.

For this reviewer, this sentence needs to be reviewed and correctly contextualized. Chagas disease was already a public health problem and was already considered a neglected disease even before it reached non-tropical regions as a result of forced migration due to violence and poverty suffered in several Latin American countries. Please, consider reviewing and rewriting that sentence.

Introduction page 2. L. amazonensis instead of L. amazonenses

Introduction, last paragraph. Authors are encouraged to make clearer to the readers the novelty of this study because the role of iron overload or depletion in the growth rate and pathogenicity of the parasite has been demonstrated since early studies in the 1980s.

M&M page 2, section 2.1. Clearly indicate why the Dm28c strain was used in this study and report in which biological culture collection the used strain is deposited. Also provide details on the iron-depleted medium; do not limit the information to citing your own reference but provide details about IDM preparation. In the discussion section, authors are encouraged to briefly elaborate on the influence of the strain used (Dm28c ) in the results observed.

M&M page 2, section 2.3. Important information on qPCR-based expression analysis is lacking. Please include information on the qPCR conditions, as well as on the amplification efficiencies and correlation coefficients (R2) obtained for primers used to amplify each gene of interest. Also provide information on the method used for quantifying gene expression. Besides primer sequences, also provide information on the amplicon sizes and accession numbers of each gene used for qPCR experiments.

Section 2.15. Please specify is SOD activity is specific (SODA, SODB) or total. In the discussion section, the authors sentenced “Fe and heme depletion leads to a downregulation of mitochondrial FeSOD (FeSODA)”; however, this reviewer could not observe, in the results section, data that support that sentence. Please moderate that statement or clarify.

Figure 6c. Please provide a better Western Blot image and correct the caption of the title of y axis (GAPDH relative abundance). Also correct in the legend western blotting instead of immunoblotting (according to M&M, authors used purified antibodies, not sera).

Figure 1. How is the increased expression of Fe reductase (TcFR) explained without increased expression of the Fe transporter (TcIT) in Fe-repleted parasites? Please include a brief explanation in the discussion section.

Please provide better (and larger) figures for the figure 9. The ones provided are at low resolution.

Author Response

Reviewer 3.

General comment.

Citation: “The manuscript antioxidants-2320474 reports a comprehensive analysis on the effect of iron depletion and repletion in energetic metabolic pathways, intracellular ROS formation, and proliferation and differentiation of epimastigotes of Trypanosoma cruzi. The description of metabolic adaptations, among other results, is an important advance to improve our understanding of the parasite’ plasticity. Technically, the study is sound, and the different methodologies used for answering the questions are fair. Authors are  encouraged to address some few issues to improve the manuscript.

Answer: Thank you for your initial comment. The suggestions and criticisms have been addressed throughout the RM.

Specific comments.

Comment 1. Citation: “Introduction page 1. “ However, several cases have lately been reported in Europe, North America, and Japan. As a result, Chagas disease became a significant public health concern worldwide [3]”. For this reviewer, this sentence needs to be reviewed and correctly contextualized. Chagas disease was already a public health problem and was already considered a neglected disease even before it reached non-tropical regions as a result of forced migration due to violence and poverty suffered in several Latin American countries. Please, consider reviewing and rewriting that sentence.

Answer: This sentence has been modified, following the recommendation by the Reviewer (page 2, line 42).

Comment 2.

Citation: “Introduction page 2. L. amazonensis instead of L. amazonenses”.

Answer: The correction has been made and the parasite is mentioned as L. amazonensis throughout the RM.

Comment 3.

Citation: “Introduction, last paragraph. Authors are encouraged to make clearer to the readers the novelty of this study because the role of iron overload or depletion in the growth rate and pathogenicity of the parasite has been demonstrated since early studies in the 1980s.

Answer: At the end of the Introduction section, we clearly state that this study describes molecular mechanisms and intracellular processes modulated by exogenous ionic Fe not previously reported (page 3, lines 103 to 104).

Comment  4.

Citation: “M&M page 2, section 2.1. Clearly indicate why the Dm28c strain was used in this study and report in which biological culture collection the used strain is deposited. Also provide details on the iron-depleted medium; do not limit the information to citing your own reference but provide details about IDM preparation. In the discussion section, authors are encouraged to briefly elaborate on the influence of the strain used (Dm28c ) in the results
observed.”

Answer: This comment is answered in three parts.

1) We used the Dm28c strain for two reasons: it differentiates “in vitro” and it was previously used in our previous studies that demonstrated the existence of a Fe-reductase and a Fe-transporter in T. cruzi [26,30]. This strain is deposited in the Collection of Trypanosoma from Wild and Domestic Mammals and Vectors (COLTRYP), Oswaldo Cruz Foundation, Rio de Janeiro, Brazil. These informations are given on page 3, lines 124 to 127 (RM).

2) We now provide details regarding the preparation of IDM, as requested (page 3, lines 109 to 119; RM

3) With respect to the third part of the comment, we mention in the Discussion section that using the Dm28c strain, which differentiates to trypomastigotes, allowed us to investigate the influence of exogenous Fe on the evolution of epimastigotes to trypomastigotes and, therefore, on a vital step of the parasite´s life cycle (page 14, line 558 to page 15, line 561; RM).

Comment 5.

Citation: “M&M page 2, section 2.3. Important information on qPCR-based expression analysis is lacking. Please include information on the qPCR conditions, as well as on the amplification efficiencies and correlation coefficients (R2) obtained for primers used to amplify each gene of interest. Also provide information on the method used for quantifying gene expression. Besides primer sequences, also provide information on the amplicon sizes and accession numbers of each gene used for qPCR experiments.

Answer: Section 2.3 has been totally rewritten to provide the information requested by the Reviewer (page 4, lines 160 to 163; RM), which is complemented in Supplementary Table 1 (Primers informations). Two new proper references (numbers 38 and 39) were added in section 2.3. Therefore, the numbering of references changed from the previous reference number 38 onwards.

Comment 6.

Citation: “Section 2.15. Please specify is SOD activity is specific (SODA, SODB) or total. In the discussion section, the authors sentenced “Fe and heme depletion leads to a downregulation of mitochondrial FeSOD (FeSODA)”; however, this reviewer could not observe, in the results section, data that support that sentence. Please moderate that statement or clarify.

Answer: Thank you for this comment. The SOD activity measured was total, as stated in RM section 2.15 (page 7, line 282; RM). For this reason, we modified the statement on the Discussion section (page 16, lines 624 to 627; RM).
Comment 7.

Citation: “Figure 6c. Please provide a better Western Blot image and correct the caption of the title of y axis (GAPDH relative abundance). Also correct in the legend western blotting instead of immunoblotting (according to M&M, authors used purified antibodies, not sera).”

Answer: We now provide a better Western blotting image in this figure (Figure 4C, page 13 in RM) and modified the legend of the “y” axis: GAPDH relative abundance instead of GAPDH (page 13; RM). We also corrected the legend of this figure: we use Western blotting instead of immunoblotting (page 13, line 494; RM).
Comment 8.

Citation: “Figure 1. How is the increased expression of Fe reductase (TcFR) explained without increased expression of the Fe transporter (TcIT) in Fe-repleted parasites? Please include a brief explanation in the discussion section.

Answer: Many thanks for this interesting comment. Notably, in the IDM+Fe medium, the expression of TcFR increased without a parallel increase in TcIT transcription. Even though TcFR and TcIT are coupled in the process of Fe uptake by the parasite, their expression is differentially modulated by the intracellular Fe content. Free Fe could regulate the TcIT transcription, so in the IDM+Fe medium, TcIT is downregulated relative to the TcIT transcript in IDM. Differently, TcFR uses Fe-containing proteins for the reaction Fe3+®Fe2+, and since these proteins (hemin and transferrin) were neither added to IDM nor to IDM+Fe, TcFR is upregulated in both cases (Table 2). This proposal has been incorporated in a new paragraph (page 15, lines 562 to 570; RM).

Comment 9.

Citation: “Please provide better (and larger) figures for the figure 9. The ones provided are at low resolution.”

Answer: The original panels A and B in Figure 9 (Figure 6 in the RM) are now larger and presented at high resolution, as requested (page 17; RM).

Reviewer 4 Report

Abstract: it is inappropriate. According to instruction for authors, four main paragraphs without headings should be included, i.e. background, methods, results and conclusions. In current work only background is described. delete “friend and foe”

 References: When citing in square brackets, reference sources should be separated by a cable, not a semicolon.

 General remarks: Based on general rules, at the beginning of sentence full species names should written, so Trypanosoma cruzi, instead of T. cruzi

 Introduction: it is too long, too many aspects are described. Authors should reduce paragraphs and sentences, that the main aspects directly related to the current research would be highlighted

 Table 1: from the table it is not clear how many genes were analyzed. “Sequence name” it is not clear, primer name?

 Material and Methods: procedures are well described; however it is not clear how many samples, repetitions were used. For instance transmission electron microscopy analysis, how many samples were examined? Improve sentence “GraphPad Prism 7.0 was used”. Statistics: SEM abbreviation should be explained.

 Figure 3: it is not clarified in the figure caption what is "M"? Authors should describe in the text alterations in details, what is specifically changed, which organelles, or general appearance of epimastigotes, such data is relevant for further discussion and investigations.

 3.4 T. cruzi have to be in italic

 Figure 9: last sentence should be: The figure was designed using BioRender.com.

Author Response

Reviewer 4.

Comment 1.

Citation: “Abstract: it is inappropriate. According to instruction for authors, four main paragraphs without headings should be included, i.e. background, methods, results and conclusions. In current work only background is described. delete “friend and foe.

Answer: The abstract has been totally rewritten in the RM, which now presents four main paragraphs (background, methods, results and conclusion) without heading, as requested (page 1, line 16 to page 2, line 35; RM).

Comment 2.

Citation: “References: When citing in square brackets, reference sources should be separated by a cable, not a semicolon.

Answer: The corrections have been made throughout the RM and are highlighted in the marked version.

Comment 3.

Citation: “General remarks: Based on general rules, at the beginning of sentence full species names should written, so Trypanosoma cruzi, instead of T. cruzi.

Answer: Thank you for your comment. Now, at the beginning of the sentences, it was used the full species name throughout the RM.

Comment 4.

Citation: “Introduction: it is too long, too many aspects are described. Authors should reduce paragraphs and sentences, that the main aspects directly related to the current research would be highlighted.”

Answer: Several paragraphs and sentences have been reduced in the Introduction of RM. For example, the redox reaction catalyzed by Fe were deleted, and the first sentence was reduced, as recommend by Reviewer 3 in his/her comment 1. We added a short sentence briefly describing the aim of the study (page 3, lines 101 to 103) (please see comment by Reviewer 2).

Comment 5.

Citation: “Table 1: from the table it is not clear how many genes were analyzed. “Sequence name” it is not clear, primer name?

Answer: Table 1 is now modified, and it is stated “Primers name” instead of “Sequence name”. In the title of the Table, we indicate that 9 genes were analyzed (page 4, line 164).
Comment 6.

Citation: “Material and Methods: procedures are well described; however it is not clear how many samples, repetitions were used. For instance transmission electron microscopy analysis, how many samples were examined? Improve sentence “GraphPad Prism 7.0 was used. Statistics: SEM abbreviation should be explained.”

Answer: In the RM we clearly state that GraphPad Prism 7.0 was used for statistics analysis and for the preparation of figures (page 7, lines 314 to 315; RM). The SEM abbreviation is explained in the text (page 7, line 305; RM) and in the list of abbreviations (page 19, line 739; RM). The number of culture samples is given in the legends of Figure 2 (page 9, line 366; RM) and Figure 3 (page 10, line 391; RM).

Comment 7.

Citation: “Figure 3: it is not clarified in the figure caption what is "M"? Authors should describe in the text alterations in details, what is specifically changed, which organelles, or general appearance of epimastigotes, such data is relevant for further discussion and investigations.

Answer: The modified legend of Figure 3, clearly mention that “M” corresponds to mitochondria (page 10, line 387).

Comment 8.

Citation: “3.4 T. cruzi have to be in italic.”

Answer: The correction has been done (page 11, line 420; RM).

Comment 9.

Citation: “Figure 9: last sentence should be: The figure was designed using BioRender.com.”

Answer: The last sentence of the legend was modified as requested by the Reviewer (page 18, line 683; RM).

Round 2

Reviewer 2 Report

It has been improved following the comments.